# Policy stringency during the COVID-19 pandemic and healthcare services utilization in China: An interrupted time-series analysis

Hong Xiao[1]*, Guannan Bai[2], Fang Liu[3], Yuechong Cui[4,5]*, Joseph M. Unger[1]

**1** Public Health Sciences Division, Fred Hutch Cancer Center, Seattle, Washington, United States of America, **2** Children's Hospital, Zhejiang University School of Medicine, National Clinical Research Center for Child and Adolescents' Health and Diseases, Hangzhou, Zhejiang, China, **3** Independent Consultant, Beijing, China, **4** Yiwu Maternity and Children Hospital (Yiwu Hospital of Children's Hospital Zhejiang University School of Medicine), Yiwu, Zhejiang, China, **5** Department of Children's and Adolescent Health, Public Health College of Harbin Medical University, Harbin, Heilongjiang, China

\* xiaohongpku@gmail.com, hxiao2@fredhutch.org (HX); cuiyuechong@163.com (YC)

## Abstract

### Background

The COVID-19 pandemic has profoundly impacted healthcare systems worldwide, with China presenting a unique case. As the first country to report COVID-19 cases and the last to lift its stringent Zero-COVID policy, China presents a distinctive context for understanding the long-term effects of the pandemic on healthcare utilization. This study provides a comprehensive analysis of healthcare utilization trends in China over more than four years of the pandemic, focusing on how different phases, including the Zero-COVID policy and its cessation.

### Methods and findings

We conducted an interrupted time-series analysis of monthly healthcare utilization data from January 2015 to April 2024, including outpatient visits and inpatient discharges, across Mainland China, controlling for underlying secular trends and patterns. Hospital-based healthcare utilization data were sourced from the National Health Commission of China, and daily Policy Stringency Indices (higher values indicating stricter control policies) were obtained from Oxford's COVID-19 Government Response Tracker. We modeled changes in healthcare utilization using negative binomial regression, comparing actual outcomes with counterfactual estimates based on pre-COVID trends. We assessed healthcare utilization during key pandemic phases, including the post-Zero-COVID period. Healthcare utilization in China experienced substantial declines during the pandemic, with an estimated reduction of 1.21 billion (7%) outpatient visits and 140.9 million (13%) inpatient discharges compared to expected levels from January 2020 to April 2024. The most pronounced declines occurred during the initial pandemic waves and coincided with periods of stringent

**Data availability statement:** All data used in this study are publicly available (Health service data: https://www.nhc.gov.cn/mohws-bwstjxxzx/new_index.shtml; Policy Stringency Index: https://www.bsg.ox.ac.uk/research/covid-19-government-response-tracker; COVID cases: https://www.healthdata.org/research-analysis/health-topics/covid). The minimal dataset underlying the findings reported in this article, together with relevant documentation, is available in the GitHub repository: https://github.com/xiaohongpku/healthservice2015to2024.

**Funding:** Research reported in this publication was supported by the Public Health Sciences Division of the Fred Hutch Cancer Center (to JMU) and "2022 Zhejiang Province High-Level Talent Training Program - Medical Rising Star Fund Support" (to YC). The funders had no other role in study design, data collection and analysis, decision to publish, or preparation of the manuscript.

**Competing interests:** The authors have declared that no competing interests exist.

**Abbreviations:** CIs, confidence intervals; IQR, interquartile range; IRR, incidence rate ratio; OxCGRT, Oxford Coronavirus Government Response Tracker; RR, rate ratio; STROBE, Strengthening the Reporting of Observational Studies in Epidemiology.

Zero-COVID measures. Negative associations between the Policy Stringency Index and healthcare utilizations were observed. Before the lifting of the Zero-COVID policy, a 10-point increase in the Policy Stringency Index was associated with a 7.2 percentage point decrease in outpatient visits and a 6.2 percentage point decrease in hospitalizations. Although healthcare utilization gradually rebounded following the cessation of the Zero-COVID policy, as of April 2024, utilization remained below expected levels in 20 (65%) of the 31 regions for outpatient visits and in 23 (74%) for inpatient discharges. Regional disparities were evident, with more developed areas, such as Shanghai and Beijing, experiencing the largest absolute reductions after adjusting for population size. In Shanghai, outpatient visits declined by 4,997 and hospitalizations by 241 per 1,000 people. In contrast, the largest relative reductions occurred in less developed regions, where outpatient visits dropped by 16% in Guizhou and hospitalizations declined by 27% in Shanxi. Use of aggregated routine health system data limited individual-level analyses, assessment of care quality, and disentangling of causal pathways.

## Conclusions

The COVID-19 pandemic and Zero-COVID policies were associated with substantial and enduring disruptions to healthcare utilization in China, characterized by slow recovery and regional disparities in access. These findings underscore the importance of strengthening healthcare systems to enhance resilience and better balance public health interventions with the maintenance of essential healthcare services in anticipation of future public health crises. Continued targeted efforts are needed to address the delayed recovery, particularly in regions with already strained healthcare infrastructure, and to ensure equitable healthcare access across the country.

Author summary

**Why was this study done?**

- The COVID-19 pandemic was associated with major disruption to healthcare systems globally, with substantial variation in the extent of disruption and recovery across countries

- Existing studies on China have been limited to specific regions or timeframes, primarily assessing short-term association such as early outbreaks in 2020 or localized surges like the 2022 Shanghai Omicron wave.

- China's unique pandemic trajectory, characterized by prolonged Zero-COVID policies followed by an abrupt cessation, remains insufficiently studied in terms of its long-term consequences for healthcare access.

**What did the researchers do and find?**

- This study performed a comprehensive, nationwide evaluation of healthcare utilization trends in China over more than four years of the COVID-19 pandemic.

- Integrating hospital-based healthcare utilization data with policy stringency indices, it quantifies the cumulative disruption and recovery of healthcare services across 31 regions.

- Through interrupted time-series analysis, the study demonstrates that as of April 2024, healthcare utilization in most regions remains below pre-pandemic projections, with pronounced disparities between more and less developed areas.

**What do these findings mean?**

- The findings highlight persistent changes in healthcare utilization observed during periods of stringent pandemic control measures, underscoring the need for policies that balance infectious disease containment with the maintenance of essential health services.

- The slow and uneven recovery in healthcare utilization may have long-term implications for population health, particularly in regions with already limited healthcare resources.

- Future public health preparedness strategies should incorporate mechanisms to sustain essential medical services during crises, ensure equitable recovery across regions, and mitigate disruptions to healthcare access during future pandemics.

- The use of aggregated routine health system data limited individual-level analyses and assessment of care quality

## Introduction

As the world continues to emerge from the most severe phases of the COVID-19 pandemic, understanding the long-term impacts on healthcare systems becomes essential. China has faced unique challenges during the COVID-19 pandemic, being the first to report cases and to implement stringent control measures, including lockdowns. It was also the last to lift a stringent Zero-COVID policy aimed at rapidly halting transmission ("find one, end one") [1,2]. This trajectory underscores the need for a comprehensive analysis of China's experience.

The COVID-19 pandemic has revealed stark variations in how different countries have managed healthcare delivery and recovery in a public health crisis [1,3–7]. The path to restoring healthcare delivery during and after a public health emergency is shaped by multiple factors, including the nature of the crisis, the types of services and populations affected, the resilience of the healthcare system, and the control measures implemented by the government. For instance, after the 2014−2015 Ebola outbreak, most primary healthcare indicators in Liberia returned to pre-Ebola levels within a year, while Guinea experienced a slower and less complete recovery [8]. Following the COVID-19 pandemic, healthcare utilization showed a complex and uneven recovery, with some services and regions surpassing pre-pandemic levels while others remained below. In South Korea, healthcare use dropped sharply in early 2020 but largely rebounded by late 2022, with variations by age and service type [9]. In Japan, inpatient volumes remained about 8% below pre-pandemic levels through 2023 [10]. In contrast, utilization in California returned to pre-pandemic levels by early 2023 [11].

In China, the pandemic initially led to a substantial decline in healthcare utilization across all levels of health facilities nationwide [1,12–14]. Although healthcare services gradually resumed, the recovery was uneven and frequently disrupted by outbreaks and lockdowns, highlighting the complex interplay between pandemic control policies and healthcare system resilience [2].

PLOS Medicine

More than five years since the onset of COVID-19, studies about the impact of the pandemic on healthcare utilization in China have been limited in scope, focusing on early phases or isolated events, such as the Shanghai Omicron outbreak in 2022 [15–17]. China's unique trajectory—characterized by early, extensive, and stringent control measures, followed by a prolonged period of low transmission, and then a sudden surge in cases due to the lifting of the Zero-COVID policy—warrants a broader assessment of long-term changes. We comprehensively analyzed national trends in healthcare utilization over the first four years of the pandemic, examining how different phases and shifts in policy stringency were associated with access to and delivery of care across regions and service types, and exploring the relationship between policy stringency and healthcare utilization.

## Methods

### Data sources and outcomes

**Healthcare utilization.** Monthly healthcare utilization data from Chinese hospitals were obtained from the Center for Health Statistics and Information, National Health Commission of China (https://www.nhc.gov.cn/mohwsbwstjxxzx/new_index.shtml) [1]. Hospitals at all levels, including both public and private, report monthly healthcare utilization data to provincial platforms, where it is verified, aggregated, and submitted to the National Health Commission for quality control. For this study, we obtained data on monthly hospital visits (outpatient and emergency) and inpatient discharges from January 2015 to April 2024, by region (province, municipality, and autonomous region) in mainland China. Population estimates by region and year were sourced from the China Statistical Yearbooks.

**Policy stringency index.** The daily stringency index by province was obtained from the Oxford Coronavirus Government Response Tracker (OxCGRT; https://www.bsg.ox.ac.uk/research/covid-19-government-response-tracker) [18]. The index is a composite measure of nine response metrics including school closures, workplace closures, public event cancelations, public gathering restrictions, public transport closures, stay-at-home requirements, public information campaigns, internal movement restrictions, and international travel controls [19]. The index, ranging from 0 to 100, reflects the intensity of measures, with higher values indicating stricter policies. Data for January 1, 2020, to February 28, 2023, were accessed on May 20, 2024.

**COVID-19 cases.** We obtained confirmed daily COVID-19 case numbers by province in China from January 1, 2020, to November 30, 2022, from the OxCGRT. Following the cessation of daily COVID data reporting by the Chinese government during the nationwide Omicron outbreak in December 2022, we used daily case estimates for December 1, 2022, to April 30, 2024, provided by the Institute for Health Metrics and Evaluation (https://www.healthdata.org/research-analysis/health-topics/covid) [20].

The analysis of de-identified, publicly available summary data does not constitute human subjects research as defined by regulation (45 CFR 46.102[d]). Therefore, no additional ethical approval was sought for conducting the study from the authors' affiliated institutions.

### Study design and statistical analysis

This study tested the following pre-specified hypotheses: (1) was the COVID-19 pandemic associated with statistically significant changes in outpatient and inpatient healthcare utilization? (2) were stricter Zero-COVID policies, as measured by the Policy Stringency Index, associated with larger reductions in healthcare utilizations? (3) Has healthcare utilization recovered to pre-COVID levels as of April 2024? (4) Did the magnitude of changes in healthcare utilization vary across regions?

We employed an interrupted time series design to estimate changes in healthcare utilization (i.e., outpatient or inpatient volume) associated with different pandemic periods as experienced within China [21]. Using the secular trend imputed from the training data in the pre-COVID-19 period, we extrapolated expected utilization for the COVID-19 period (January

2020–April 2024) to estimate what utilization would have been in the absence of the COVID-19 pandemic. Given a dispersed variation structure in monthly counts of utilizations and likelihood ratio tests favoring negative binomial regression over Poisson equivalents ($p < 0.05$ for both aggregated national data and provincial-level data), we modeled the outcomes using negative binomial regressions. We incorporated a linear term for time to account for the long-term secular trend in healthcare utilization during the pre-COVID period (January 2015 to December 2019). Fixed-effect monthly indicators and the number of Spring Festival days were included to account for seasonal patterns and theassociationof the Spring Festival with hospital admissions. We applied Newey-West standard errors with up to three lags, per results of the PACF (S1 and S2 Figs), to address autocorrelation.

We reported the incidence rate ratio (IRR) of the observed (factual) utilization versus model-expected (counterfactual) estimates across seven prespecified period—first-wave peak (Feb–Mar 2020), recovery (Apr–Jul 2020), low-transmission (Aug 2020–Mar 2022), Shanghai Omicron outbreak (Apr–May 2022), Omicron wave (Jun–Nov 2022), lifting of Zero-COVID (Dec 2022–Jan 2023), and post–Zero-COVID (Feb 2023–Apr 2024)—and overall from Jan 2020 to Apr 2024. Secondarily, we calculated absolute differences in outpatient visits and inpatient discharges. We generated 1,000 monthly counterfactual predictions by sampling from the model's multivariate normal coefficient distribution (i.e., based on their asymptotic distribution); the 2.5th and 97.5th percentiles defined 95% confidence intervals (CIs), and two-sided $P$ values were computed as twice the smaller proportion of simulations above or below zero.

We performed separate regression analyses for outpatient visits and inpatient volumes at both the national and regional levels. We assessed the number and proportion of regions where healthcare utilization had not returned to expected levels by April 2024, considering utilization below expected if it remained lower for three consecutive months (Feb–Apr 2024). Additionally, we used linear regression to examine the relationship between the monthly average Policy Stringency Index and relative changes in healthcare utilization.

Most missing province-level monthly utilization data occurred in December due to administrative and reporting reasons were imputed as the mean of the preceding and following months; December 2022 was imputed from January 2023 due to the abrupt end of the Zero-COVID policy. Daily provincial stringency indices after February 28, 2023, were imputed by decrementing 0.3 points per day until reaching zero, based on the empirical distribution provincial index values and the subsequent timeline of key policy relaxation timeline [22–24].

Analyses were conducted in R-version 4.0.2. Statistical significance was set at alpha = 0.05 and all tests were two-sided. This study is reported as per the Strengthening the Reporting of Observational Studies in Epidemiology (STROBE) guideline (S1 STROBE Checklist). Although this observational study was not formally preregistered, all reported analyses were planned a priori based on the study objectives, available data, and the interrupted time-series design. No analyses were added in response to unanticipated patterns in the data (i.e., no data-driven/post hoc analyses were conducted), and no material deviations from the initial analysis plan occurred. Any minor technical refinements (e.g., model specification or sensitivity checks) are explicitly noted in the Methods section along with rationale.

## Results

The number of hospitals included in the analysis increased from 26, 047 in 2015 to 38, 355 in 2024, serving a total population of approximately 1.4 billion. Increasing trends in monthly hospital outpatient visits and inpatient volume were observed across the pre-COVID period at both the national and provincial level (Figs 1 and 2).

From January 1, 2015, to April 30, 2024, there were approximately 33.8 billion outpatient visits and 1.8 billion inpatient discharges, with 17.1 billion (50.4%) outpatient visits and 0.91 billion (50.2%) inpatient discharges occurring during the COVID-19 period (Tables 1 and S1). A total of 91.4 million COVID-19 cases were reported in the study period, with approximately 98% occurring after the lifting of the Zero-COVID policy (Fig 3A). The mean provincial-level daily Stringency Index prior to the lifting of the Zero-COVID policy was 53.4 (median, 52.8; interquartile range [IQR], 17.6)). Trends in the

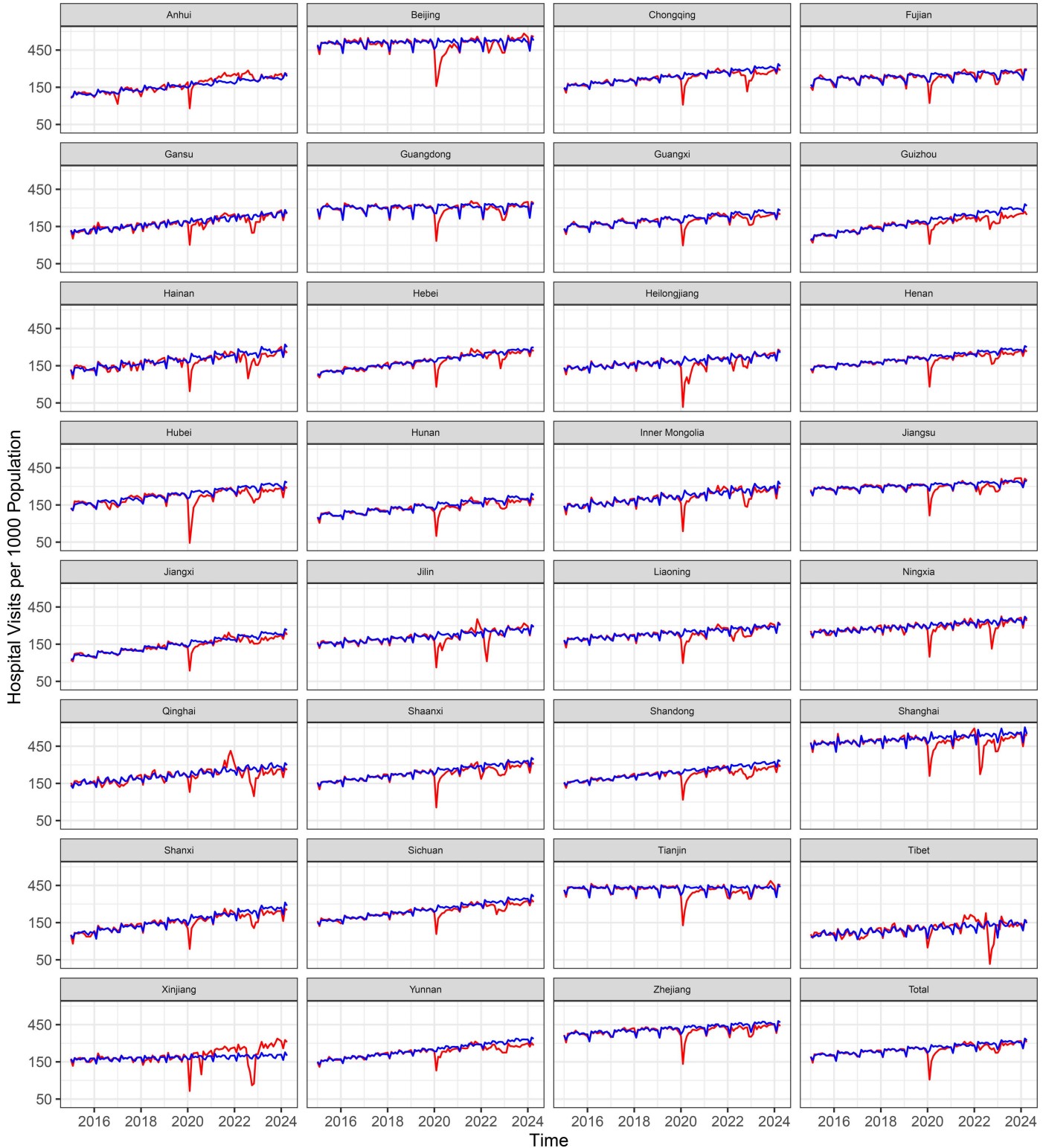

**Fig 1. Trends of observed and expected outpatient visits per 1,000 population per month.** The red lines represent observed trends, and blue lines represent expected levels, both before (January 2015–December 2019) and after (January 2020–April 2024) the onset of the pandemic.

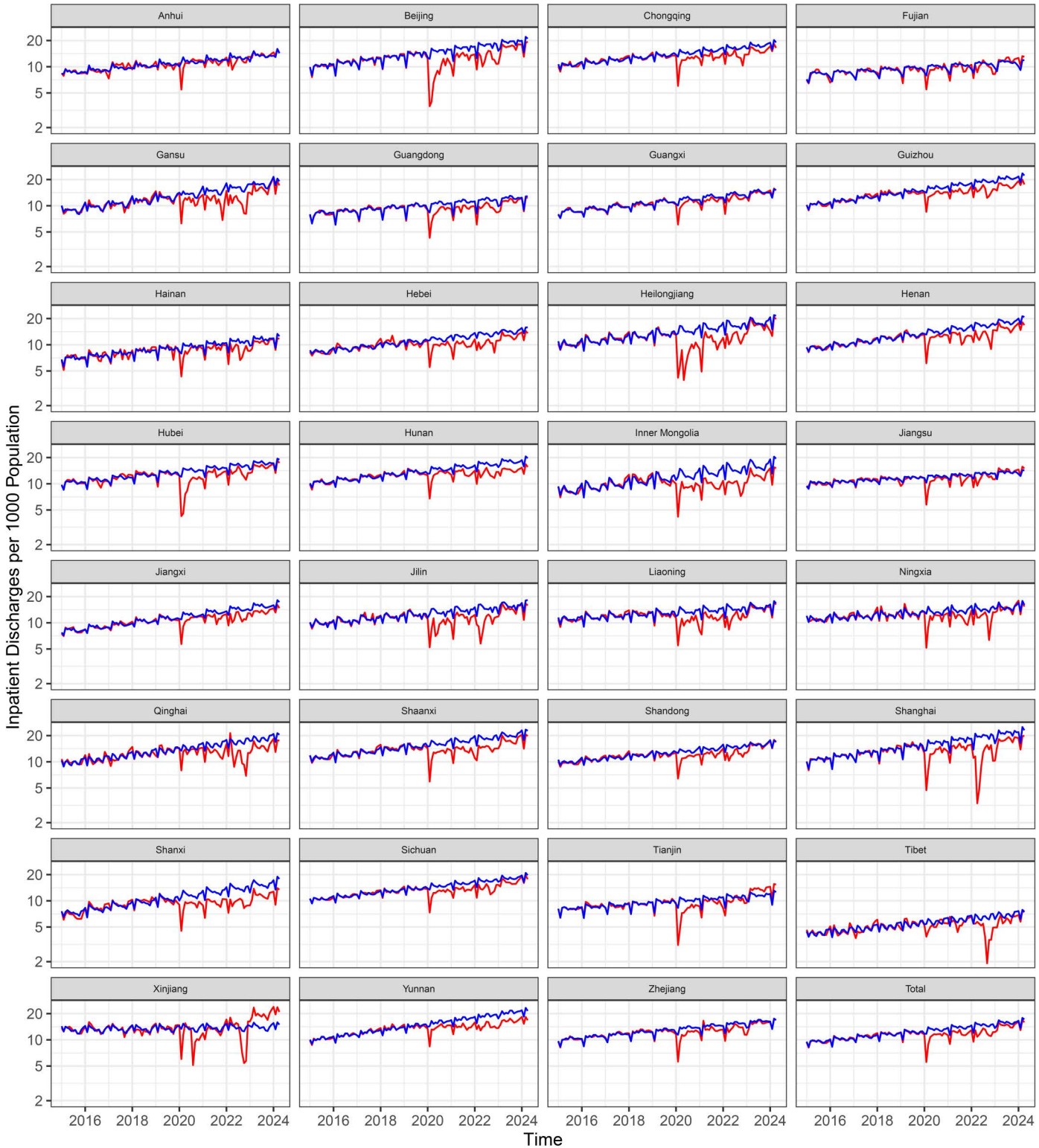

**Fig 2. Trends of observed and expected inpatient discharges per 1,000 population per month.** The red lines represent observed trends, and blue lines represent expected levels, both before (January 2015–December 2019) and after (January 2020–April 2024) the onset of the pandemic.

**Table 1. Hospital-based Healthcare utilizations, COVID-19 cases and Policy Stringency Index by year.**

| Year | Outpatient visits (in millions) | Inpatients Discharged (in millions) | COVID-19 Cases | Mean Policy Stringency Index |
|---|---|---|---|---|
| 2015 | 2,994.8 | 154.9 | 0 | 0 |
| 2016 | 3,161.9 | 168.7 | 0 | 0 |
| 2017 | 3,361.9 | 183.1 | 0 | 0 |
| 2018 | 3,541.2 | 196.4 | 0 | 0 |
| 2019 | 3,713.4 | 202.8 | 0 | 0 |
| 2020 | 3,300.1 | 180.7 | 93,132 | 53.1 |
| 2021 | 4,166.4 | 199.0 | 21,489 | 50.2 |
| 2022 | 3,851.9 | 201.8 | 54,453,659 | 60.6 |
| 2023 | 4,250.6 | 244.0 | 36,876,574 | 7.9 |
| 2024 (Jan–Apr) | 1,483.6 | 87.0 | 3,070 | 0 |
| Total | 33,825.7 | 1,818.3 | 91,447,924 | NA |

Stringency Index were broadly consistent across provinces, with peak stringency observed during the initial wave (Feb–Mar 2020) and the Shanghai Omicron outbreak (Apr–May 2022) (Fig 3B, S2 Table).

## Changes in healthcare utilization at the national level

From January 2020 to April 2024, the COVID-19 pandemic was associated with an estimated 7% loss in outpatient visits (IRR = 0.93, 95% CI [0.92, 0.95]; $p < 0.0001$) and a 13% loss in hospitalizations (IRR = 0.87, 95% CI [0.84, 0.89]; $p < 0.0001$) compared to expected levels (Table 2, Fig 4). These declines correspond to approximately 1.21 billion fewer outpatient visits (95% CI [0.86, 1.58] billion) and 140.9 million fewer hospital discharges (95% CI [111.0, 173.5] million) (Table 2). Cumulatively, this translates to 798 fewer outpatient visits and 106 fewer hospitalizations per 1,000 population over a timeframe of 4 years and 4 months, with average monthly reductions of 15 outpatient visits and 2 hospitalizations per 1,000 population (S3 and S4 Figs). The nationwide relative reductions were statistically significant across all seven COVID-19 periods for both outpatient visits and inpatient discharges. For outpatient visits, the reductions ranged from 47% (IRR = 0.53, 95% CI [0.51, 0.55]; $p < 0.0001$) at the peak of the first wave to 4% (IRR = 0.96, 95% CI [0.93, 0.98]; $p = 0.004$) in the post-Zero-COVID period. Similarly, for inpatient discharges, the reductions ranged from 44% (IRR = 0.56, 95% CI [0.54, 0.58]; $p < 0.0001$) at the peak of the first wave to 6% (IRR = 0.94, 95% CI [0.90, 0.97]; $p = 0.002$) in the post-Zero-COVID period. As of April 2024, either outpatient or hospitalization utilization had returned to the expected level (Figs 1 and 2, S3 Table)

## Changes in healthcare utilization at the subnational level

At the subnational level, overall outpatient visits decreased in 28 of the 31 regions; among these, reductions were statistically significant in all but Fujian, Hebei, and Liaoning. Inpatient discharges declined in 30 of the 31 regions, with statistically significant reductions in 28, except in Anhui and Fujian (Table 2). In contrast, outpatient visits increased in three regions (Anhui, Tibet, and Xinjiang), with statistically significant increases in Anhui ($p = 0.01$) and Xinjiang ($p < 0.0001$). Inpatient discharges increased only in Xinjiang, although the increase was not statistically significant ($p = 0.16$). The largest relative reduction in outpatient visits was observed in Hubei during the first wave peak, with a 69% decline (rate ratio [RR], 0.31; 95% CI [0.29, 0.34]; $p < 0.001$) (S4 Table). The largest reduction in inpatient discharges occurred in Shanghai during the Omicron outbreak, with a 79% decline between April and May 2022 (RR = 0.21; 95% CI [0.20, 0.22]; $p < 0.001$) (S5 Table).

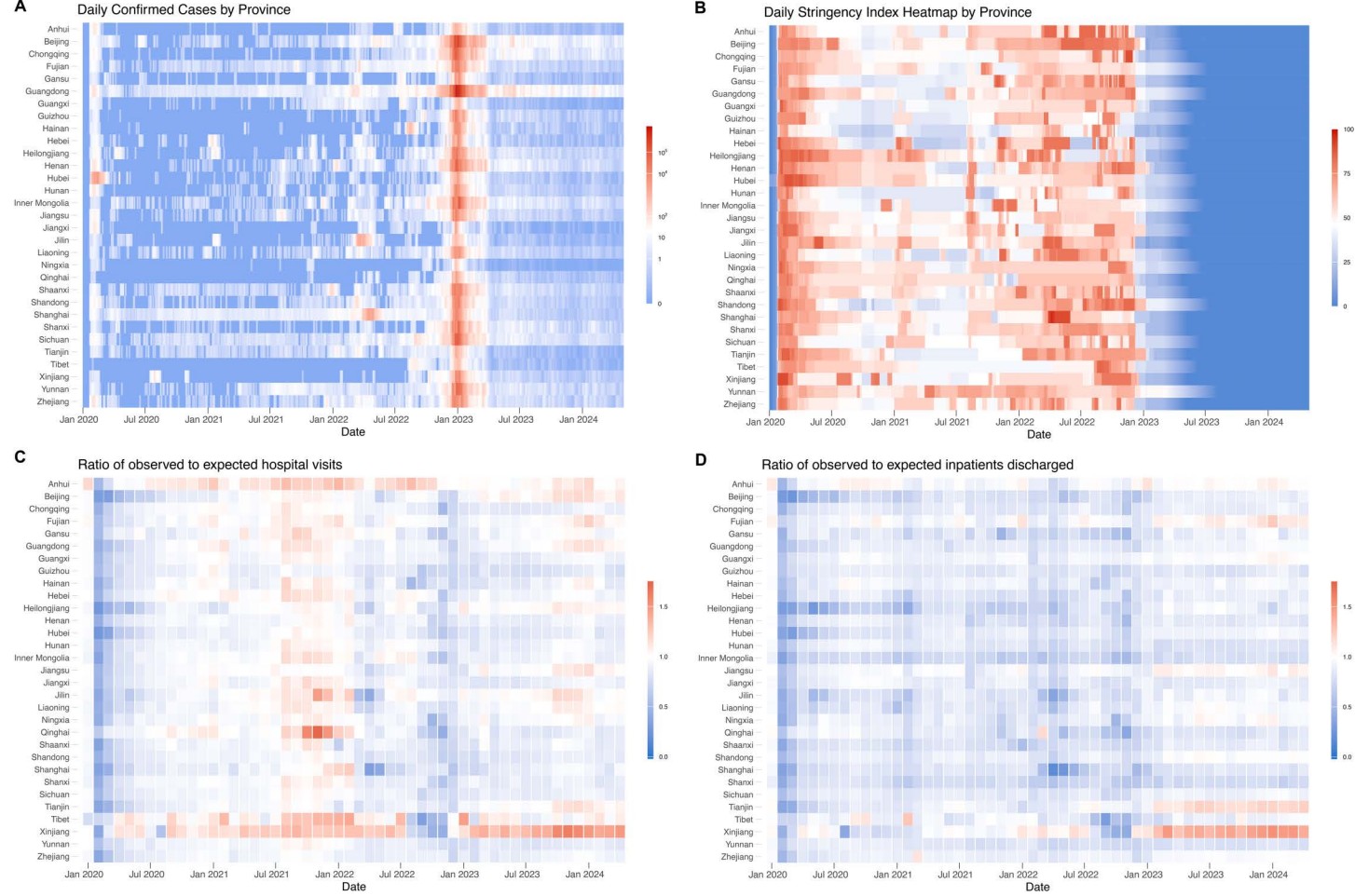

**Fig 3. Daily confirmed cases, Policy Stringency Index and monthly ratio of observed to expected healthcare utilization by regions.** Panel **A**: Daily confirmed cases by region over time; Panel **B**: Daily policy stringency index by region over time; Panel **C**: Ratio of observed to expected monthly hospital visits by region over time; Panel **D**: Ratio of observed to expected monthly inpatient discharges by region over time. In all panels, red indicates high values and blue indicates low values.

As of April 2024, Shanghai experienced the highest absolute cumulative loss in outpatient visits, with a reduction of 4,997 (95% CI [3,710, 6,360]) visits per 1,000 population, followed by Zhejiang, with a reduction of 2,839 (95% CI [1,920, 3,911]) visits per 1,000 population (S3 Fig). The largest relative cumulative decreases were observed in Guizhou (RR = 0.84, 95% CI [0.79, 0.88]) and Shanghai (RR = 0.84, 95% CI [0.81, 0.88]) (Table 2, S5 Fig). For inpatient discharges, Shanghai experienced the largest absolute cumulative loss, with a reduction of 241 (95% CI [203, 287]) hospitalizations per 1,000 population followed by Beijing (205;95% CI [187, 227] per 1,000 population) and Heilongjiang (203; 95% CI [177, 229] per 1,000 population) (S4 Fig). The most profound relative cumulative decreases in hospitalizations were observed in Jiangsu (RR = 0.72; 95% CI [0.65, 0.80]), Shanxi (RR = 0.73; 95% CI [0.66, 0.81]), and Shanghai (RR = 0.75; 95% CI [0.72, 0.78]) (Table 2, S5 Fig).

As of April 2024, outpatient visits had not resumed to the expected level in 20 out of the 31 regions (65%), and hospitalization remained below the expected level in 23 (74%) regions (S3 Table).

**Table 2. Overall changes in outpatient visits and hospitalizations by regions from January 2020 to April 2024.**

| Region | Outpatient Visits | | | Inpatients Discharged | | |
|---|---|---|---|---|---|---|
| | IRR (95% CI) | Difference (95% CI)* | P-value | IRR (95% CI) | Difference (95% CI)* | P-value |
| Anhui | 1.09 (1.02, 1.16) | 51.45 (12.71, 88.76) | **0.01** | 0.96 (0.91, 1.03) | −1.40 (−3.88, 1.19) | 0.29 |
| Beijing | 0.93 (0.89, 0.96) | −48.64 (−73.91, −25.13) | <.0001 | 0.77 (0.75, 0.78) | −4.49 (−4.95, −4.08) | <.0001 |
| Chongqing | 0.91 (0.88, 0.94) | −37.02 (−50.68, −24.55) | <.0001 | 0.85 (0.81, 0.88) | −4.10 (−5.23, −3.01) | <.0001 |
| Fujian | 0.97 (0.94, 1.01) | −12.80 (−31.79, 5.80) | 0.17 | 0.98 (0.93, 1.02) | −0.56 (−1.66, 0.52) | 0.31 |
| Gansu | 0.96 (0.92, 0.99) | −10.88 (−20.19, −1.64) | 0.02 | 0.82 (0.77, 0.86) | −3.73 (−4.88, −2.65) | <.0001 |
| Guangdong | 0.95 (0.92, 0.98) | −90.02 (−149.28, −32.01) | <.0001 | 0.86 (0.84, 0.88) | −10.53 (−12.39, −8.75) | <.0001 |
| Guangxi | 0.91 (0.88, 0.94) | −50.70 (−67.44, −33.51) | <.0001 | 0.93 (0.91, 0.96) | −2.32 (−3.28, −1.40) | <.0001 |
| Guizhou | 0.84 (0.79, 0.88) | −73.24 (−95.57, −50.50) | <.0001 | 0.83 (0.78, 0.88) | −6.40 (−8.76, −4.26) | <.0001 |
| Hainan | 0.87 (0.82, 0.92) | −15.09 (−21.72, −9.07) | <.0001 | 0.86 (0.81, 0.91) | −0.81 (−1.13, −0.50) | <.0001 |
| Hebei | 0.96 (0.92, 1.01) | −30.44 (−66.28, 5.89) | 0.09 | 0.83 (0.76, 0.91) | −8.60 (−13.02, −4.29) | <.0001 |
| Heilongjiang | 0.94 (0.92, 0.97) | −16.96 (−24.43, −8.58) | <.0001 | 0.76 (0.74, 0.79) | −6.22 (−7.01, −5.42) | <.0001 |
| Henan | 0.91 (0.89, 0.93) | −101.12 (−126.54, −75.25) | <.0001 | 0.83 (0.79, 0.85) | −14.65 (−17.95, −11.86) | <.0001 |
| Hubei | 0.85 (0.80, 0.91) | −110.15 (−160.39, −63.00) | <.0001 | 0.84 (0.79, 0.88) | −7.79 (−10.51, −5.20) | <.0001 |
| Hunan | 0.93 (0.89, 0.96) | −42.96 (−64.90, −21.80) | <.0001 | 0.83 (0.79, 0.86) | −9.74 (−11.88, −7.48) | <.0001 |
| Jiangsu | 0.93 (0.89, 0.97) | −19.99 (−31.28, −8.15) | <.0001 | 0.72 (0.65, 0.80) | −5.09 (−7.01, −3.26) | <.0001 |
| Jiangxi | 0.95 (0.91, 0.99) | −68.24 (−122.43, −16.02) | <.0001 | 0.94 (0.90, 1.00) | −3.08 (−6.03, −0.18) | 0.04 |
| Jilin | 0.92 (0.87, 0.96) | −36.11 (−57.89, −15.93) | <.0001 | 0.86 (0.83, 0.89) | −4.55 (−5.81, −3.42) | <.0001 |
| Liaoning | 0.99 (0.95, 1.03) | −3.41 (−14.55, 7.57) | 0.56 | 0.81 (0.79, 0.84) | −3.27 (−3.74, −2.78) | <.0001 |
| Inner Mongolia | 0.95 (0.92, 0.98) | −25.68 (−41.41, −10.83) | <.0001 | 0.85 (0.80, 0.91) | −4.72 (−6.71, −2.67) | <.0001 |
| Ningxia | 0.92 (0.88, 0.97) | −8.37 (−13.24, −3.13) | <.0001 | 0.87 (0.82, 0.91) | −0.73 (−1.00, −0.46) | <.0001 |
| Qinghai | 0.96 (0.89, 1.02) | −3.14 (−8.38, 1.36) | 0.17 | 0.81 (0.76, 0.86) | −1.00 (−1.34, −0.70) | <.0001 |
| Shaanxi | 0.89 (0.87, 0.91) | −57.25 (−66.74, −47.87) | <.0001 | 0.80 (0.76, 0.83) | −7.67 (−9.65, −5.98) | <.0001 |
| Shandong | 0.86 (0.83, 0.89) | −177.08 (−222.44, −134.74) | <.0001 | 0.89 (0.85, 0.94) | −8.58 (−12.37, −4.71) | <.0001 |
| Shanghai | 0.84 (0.81, 0.88) | −124.27 (−158.18, −92.28) | <.0001 | 0.75 (0.72, 0.78) | −6.00 (−7.13, −5.04) | <.0001 |

*(Continued)*

**Table 2.** (Continued)

| Region | Outpatient Visits | | | Inpatients Discharged | | |
|---|---|---|---|---|---|---|
| | IRR (95% CI) | Difference (95% CI)* | P-value | IRR (95% CI) | Difference (95% CI)* | P-value |
| Shanxi | 0.90 (0.84, 0.97) | −35.79 (−62.85, −9.67) | 0.01 | 0.73 (0.66, 0.81) | −6.63 (−9.23, −4.26) | <.0001 |
| Sichuan | 0.89 (0.87, 0.92) | −123.28 (−161.68, −86.79) | <.0001 | 0.86 (0.83, 0.88) | −10.16 (−12.26, −8.19) | <.0001 |
| Tianjin | 0.93 (0.90, 0.96) | −20.78 (−31.07, −10.84) | <.0001 | 0.96 (0.94, 0.99) | −0.29 (−0.51, −0.08) | 0.016 |
| Tibet | 1.02 (0.94, 1.11) | 0.42 (−1.54, 2.53) | 0.66 | 0.86 (0.81, 0.92) | −0.16 (−0.25, −0.09) | <.0001 |
| Xinjiang | 1.17 (1.12, 1.23) | 41.17 (29.77, 52.21) | <.0001 | 1.04 (0.98, 1.11) | 0.80 (−0.36, 1.92) | 0.16 |
| Yunnan | 0.89 (0.87, 0.91) | −68.24 (−85.20, −51.51) | <.0001 | 0.81 (0.78, 0.85) | −8.32 (−10.26, −6.26) | <.0001 |
| Zhejiang | 0.88 (0.84, 0.91) | −188.16 (−252.52, −127.21) | <.0001 | 0.89 (0.85, 0.93) | −5.62 (−7.76, −3.48) | <.0001 |
| Total | 0.93 (0.92, 0.95) | −1214.00 (−1576.00, −864.18) | <.0001 | 0.87 (0.84, 0.89) | −140.89 (−173.48, −111.02) | <.0001 |

Note: Blue cells indicate a statistically significant decrease, while light blue cells represent a decrease that is not statistically significant; Orange cells indicate a statistically significant increase, while light orange cells represent an increase that is not statistically significant.

*Measured in millions.

### Association between the Policy Stringency Index and the change in healthcare utilization

Negative associations between the Policy Stringency Index and healthcare utilizations were observed (Fig 5). Before the lifting of the Zero-COVID policy, a 10-point increase in the Policy Stringency Index was associated with a 7.2 percentage point decrease in outpatient visits (95% CI [6.4, 7.9]; $p < 0.0001$) and a 6.2 percentage point decrease in hospitalizations (95% CI [5.6, 6.8]; $p < 0.0001$).

### Discussion

This comprehensive time-series analysis offers an in-depth assessment of changes in healthcare utilization in China during the COVID-19 pandemic from January 2020 to April 2024.. Our study reveals that healthcare utilization in mainland China was substantially disrupted during the COVID-19 pandemic, with an overall reduction of 1.21 billion (7%) outpatient visits and 140.9 million (13%) inpatient discharges compared to expected levels. Declines were most pronounced during the initial waves but persisted into the post–Zero-COVID period, with substantial regional variation. Stringent Zero-COVID policies were strongly associated with reduced healthcare utilization, while the subsequent lifting of these policies led to a temporary surge in COVID-19 cases and further disruptions. Despite gradual recovery, utilization remains below expected levels nationwide, illustrating the enduring gaps in healthcare service use in China during the study period. Strains on patients, healthcare providers, healthcare facilities, and health systems imposed by the COVID-19 pandemic have been widely reported globally and both in China [3,7,9,25–30]. The decline in health facility visits, particularly during the early waves of the pandemic, was attributable to multiple interconnected factors, including changes in health-seeking behavior, reduced demand, constrained health service delivery, and limited physical accessibility. Fear of infection and uncertainty about the virus led many patients and families to delay or forgo in-person care, even after lockdowns were lifted [31], with some substituting in-person visits with telehealth consultations [32]. Healthcare demand also declined due to a reduction

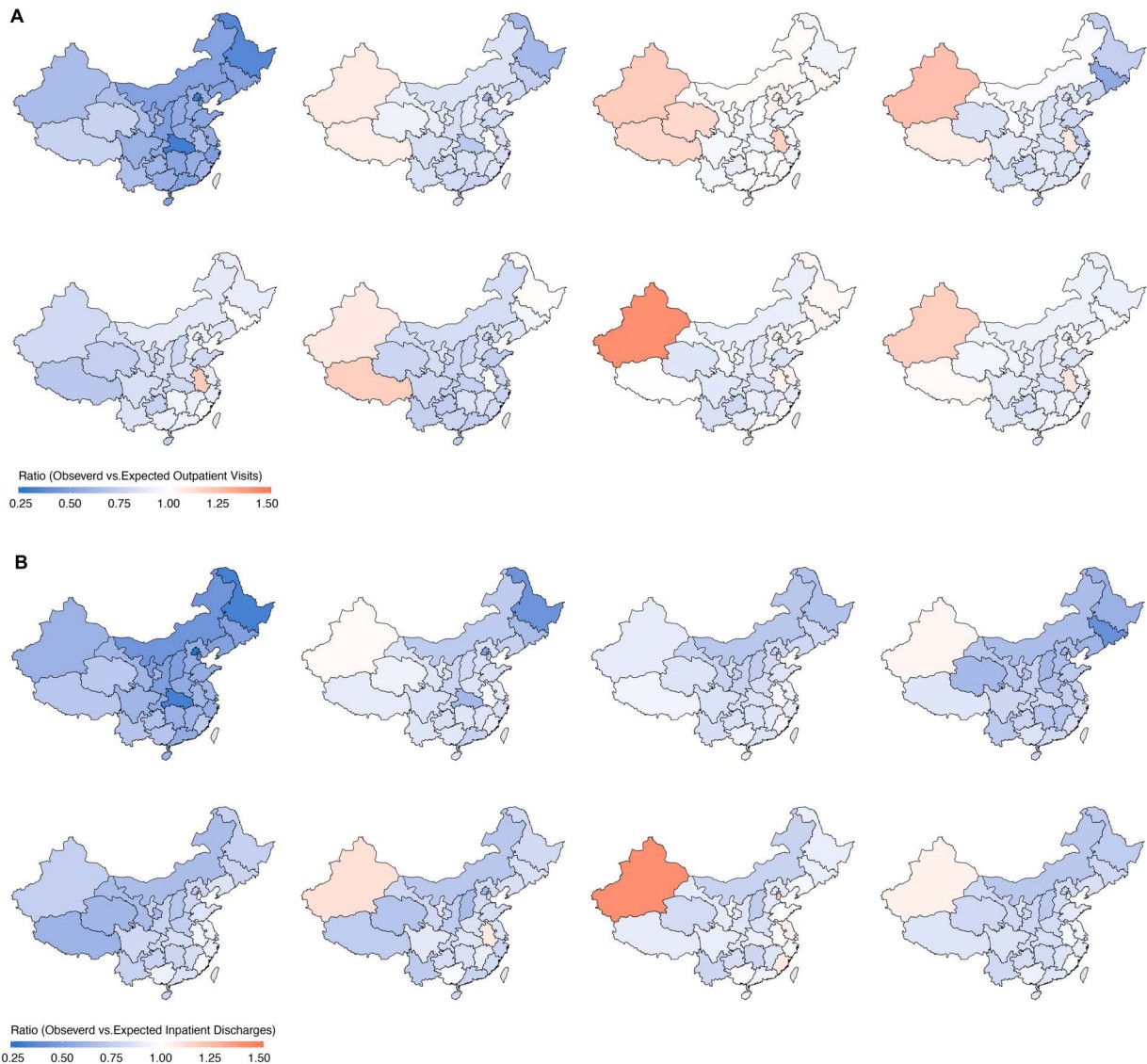

**Fig 4. Ratios of model-based observed to expected healthcare utilization during different pandemic periods.** Panel **A** represents for outpatient visits and the panel **B** illustrates inpatient discharges. The periods depicted are as follows(top left to bottom right): (1) The peak of the first wave (February 2020–March 2020); (2) The recovery period (April 2020–July 2020); (3) The period with low COVID transmission in China (August 2020–March 2022); (4) The Shanghai Outbreak (April 2020–May 2022); (5) The Omicron wave (June 2022–November 2022); (6) The lifting of Zero-COVID policy (December 2022–January 2023); (7) Post Zero-COVID period (February 2023–April 2024); (8) The entire COVID-19 period (January 2020–April 2024). Maps were generated in R using provincial boundary shapefiles obtained from GitCode (accessed February 25, 2026; MIT License). Available from: https://gitcode.com/open-source-toolkit/19fe0.

in non-COVID-19 conditions (e.g., cardiovascular diseases, influenza, injuries, and sexually transmitted infections), driven by widespread non-pharmacological interventions to contain the virus [33]. On the supply side, many health facilities—particularly outpatient departments and those providing non-emergency or chronic care—faced disruptions from service suspensions, workforce reallocation, and deliberate delays in elective procedures to minimize nosocomial transmission [34,35]. Additionally, frequent and widespread mobility restrictions, both inter- and intra-city, and suspension of public transportation substantially limited physical access to healthcare [1,36]. Collectively, these interrelated, multifaceted

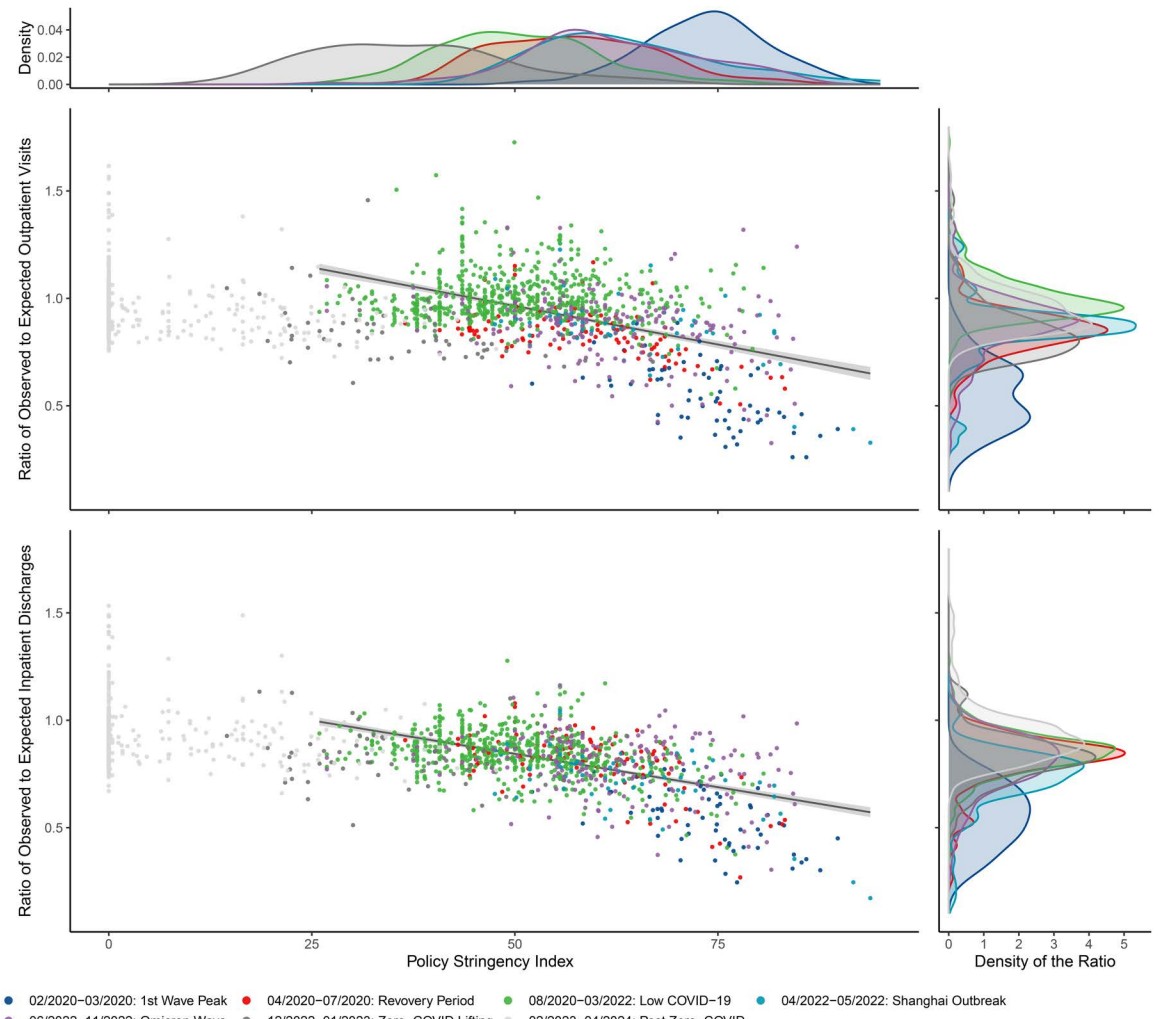

**Fig 5. Association between Policy Stringency Index (monthly average) and changes in healthcare utilizations before the lifting of the Zero-COVID policy.** The top and right panels present the distributions of the Policy Stringency Index and the monthly observed-to-expected visit ratio by phase, respectively. The bottom-left panels show a negative association between policy stringency and the observed-to-expected visit ratio.

factors contributed to the persistent disparities between actual and expected healthcare utilization observed during the pandemic.

Our findings of reduced healthcare utilization align with previous studies documenting short-term disruptions during the initial waves of the pandemic and specific outbreaks [9,17]. However, this study extends prior work by demonstrating the long-lasting association of the pandemic and associated control measures. Although utilization gradually rebounded in the post-pandemic phases, full recovery was not achieved in most regions of China through April of 2024. These trends mirror findings from other global settings, such as South Korea [9], where service volumes declined markedly and recovered unevenly by late 2022. However, China's prolonged and stringent Zero-COVID policy may have further delayed recovery compared to countries that adopted less restrictive measures [5,37].

The slow recovery of healthcare utilization in the post-Zero-COVID and post-pandemic period reflects the suggests persistent disruptions to China's healthcare system associated with the pandemic. Despite the lifting of restrictions in late

2022 and the World Health Organization's declaration in May 2023 that COVID-19 was no longer a public health emergency of international concern, outpatient visits and hospitalizations remained below expected levels in most regions of China. Several China-specific factors may explain this delayed rebound. The pandemic triggered widespread loss of health insurance coverage, driven by rising unemployment and declining household incomes that failed to keep pace with increasing insurance premiums [38–40]. Notably, enrollment in the New Rural Cooperative Medical Scheme dropped by 62 million between 2019 and 2023, with a sharp decline of 45 million enrollees occurring between 2022 and 2023 alone [41,42].

Although excess mortality in China remained low during the first two years of the pandemic [30,43], the abrupt termination of the Zero-COVID policy was followed by a sharp surge in all-cause mortality, particularly among individuals aged 65 years and older [26]. This sudden loss of medically vulnerable individuals may have temporarily reduced demand for intensive care services. Additionally, a marked decline in birth rates during and after the pandemic may have further contributed to reduced demand for both outpatient and inpatient care [44–46]. Continued uptake of telemedicine among healthcare professionals and patients—sustained for up to nine months after the easing of lockdowns—may have contributed to the slower return of in-person outpatient visits [47,48]. In parallel, China's recent medical insurance reform, which reallocates part of individual employee accounts to pooled public funds to enhance outpatient reimbursement, was accelerated in response to COVID-19. This policy shift aims to strengthen the primary care, improve healthcare financing efficiency and potentially reduce avoidable hospital admissions [49].

Regional disparities in healthcare utilization highlight heterogeneity in healthcare disruptions associated with the COVID-19 pandemic and related policies.. After adjusting for population size, the largest absolute declines in both outpatient and inpatient services occurred in highly developed regions such as Shanghai, Beijing, and Zhejiang—areas that faced higher case burdens and stricter control measures due to dense populations. More critically, inter- and intra-city mobility restrictions substantially hindered patients from less developed neighboring regions from accessing healthcare in these resource-rich areas, exacerbating existing inequalities in healthcare access. On the other hand, the relatively large decline in some less developed regions, such as Guizhou and Shanxi, could be due to their less resilient health systems and constrained capacity, particularly during critical periods when health workers and facilities were engaged in mass testing and implementing pharmaceutical interventions to reduce in-facility infections. Paradoxically, outpatient and inpatient utilization increased beyond expected levels in some under-resourced regions such as Tibet, Xinjiang, and Anhui. Historically, a larger proportion of patients from these regions travel to sought healthcare, particularly specialized care, in more developed neighboring cities and provinces [50]. However, travel restrictions and the associated rise in financial and time costs during the COVID-19 pandemic made cross-provincial healthcare seeking less feasible, thereby hindering this behavior. Data shows that cross-provincial hospitalizations dropped by 1.8 million (31%) from 2019 to 2020 [50]. Despite the increase in service utilization in these regions, concerns remain regarding whether their existing healthcare systems can meet local demand both in terms of quantity and quality.

These findings have potential implications for the long-term financial health of China's healthcare system, including decline in hospital revenue and health worker behavior. In China, most hospitals, including public ones, are largely self-financed through service charges and drug sales, with government subsidies accounting for only 9.7% to 11.7% of public hospital revenue in the Mideastern region, and slightly more in the northwest and north [51,52]. During the early months of the pandemic, total healthcare expenditures dropped by 37.8% compared to the same period the previous year, based on 300 million bank card transactions [15]. Amid this revenue decline, reports emerged of salary reductions for health workers [53,54]. Concerns have been raised that hospitals, facing financial strain, may have compensated by increasing the use of diagnostic tests and medical consumables. Despite an overall decrease in healthcare spending, per capita healthcare expenditures rose [15]. Shen and colleagues reported that in the first 25 weeks following the first wave in 2020, the average total expenses per patient rose by 8.7% to 16.7%, largely driven by higher laboratory test costs and medical

consumables — most of which the additional expenses were reimbursed by health insurance [55]. The authors suggested these increases reflected widespread profit-compensation practices by hospitals in response to lost admissions during the initial COVID-19 wave.

Our study has several strengths. We used nationwide administrative data covering all hospitals in China to estimate changes in hospital-based healthcare utilization at both national and regional levels. The five-year pre-pandemic baseline and over four years of follow-up allowed near real-time monitoring of system performance and detailed trend analysis. In addition, region-specific daily Policy Stringency indices enabled assessment of how the intensity of COVID-19 containment measures correlated with disruptions in healthcare use.

This study has several limitations. First, the use of aggregated data precluded analysis by individual characteristics such as sex, age, disease type, residency, insurance status, and socioeconomic factors. Second, we could not assess changes in care quality or the broader indirect changes on health outcomes, including quality of life, hospitalization, and mortality. Third, routine health system data did not capture telemedicine visits provided by hospitals or third-party platforms. Although modest in scale [49], their omission may have led to an overestimation of the overall collateral disruptions. Fourth, our study could not fully disentangle the specific pathways through which the COVID-19 pandemic affected healthcare utilization, or to quantify the relative contribution of each factor. Multiple intermediary mechanisms, including changes in healthcare supply, provider and patient behaviors, disease spectrum, and mortality patterns may have been involved, but detailed data for pathway analyses were not available. Lastly, our estimates of unmet need were based on counterfactual projections assuming no pandemic. However, actual healthcare needs may have changed due to shifts in environmental exposures, health-related behaviors, mental health, infectious disease prevalence, and demographic factors such as excess mortality among older adults and declining fertility. These shifts may bias our estimates in either direction. Future research should integrate individual-level data to better examine disparities in healthcare use, assess care quality and outcomes, evaluate the evolving role of telemedicine, and improve models of unmet need that account for demographic and epidemiological changes.

This study underscores the profound and enduring disruptions in healthcare utilization in China during the COVID-19 pandemic and periods of stringent control measures., highlighting the substantial disruptions that occurred across various regions. A critical challenge for policymakers, in China and globally, lies in balancing population-level mitigation strategies aimed at stemming the morbidity and mortality due to COVID-19 infections with the potential consequences of these measures can have on healthcare access. Our findings highlight the need for pandemic response strategies that account for these trade-offs and prioritize the continuity of essential care. Future preparedness efforts should incorporate surge capacity planning, alternative care delivery models such as telemedicine, and targeted public communication. Furthermore, region-specific policies will also be vital to address the heterogeneous pandemic related challenges on local health systems. Continued monitoring of healthcare utilization trends, coupled with targeted interventions to address unmet healthcare needs and mitigate the long-term consequences on population health, are crucial.

### Patient and public involvement

Patients and/or the public were not involved in the design, or conduct, or reporting, or dissemination plans of this research.

### Supporting information

**S1 Table. Population and Hospital-based healthcare utilization by regions.**
(DOCX)

**S2 Table. Region-specific Policy Stringency Index by different pandemic periods.**
(DOCX)

**S3 Table. The recovery of healthcare utilizations as of April 2024.**
(DOCX)

**S4 Table. Relative and absolute change in outpatient visits by regions and periods.**
(DOCX)

**S5 Table. Relative and absolute change in hospitalizations by regions and periods.**
(DOCX)

**S1 Fig. PACF Partial Autocorrelation Function (PACF) of residuals for outpatient visits by region.**
(PDF)

**S2 Fig. PACF Partial Autocorrelation Function (PACF) of residuals for inpatient discharges by region.**
(PDF)

**S3 Fig. Cumulative loss in outpatient utilization by region (January 2020–April 2024).** Cumulative loss is measured in number of visits per 1000-person (blue lines) and number of visits per 1000-person-month (red lines).
(PDF)

**S4 Fig. Cumulative loss in inpatient discharges by region (January 2020–April 2024).** Cumulative loss is measured in number of visits per 1000-person (blue lines) and number of visits per 1000-person-month (red lines).
(PDF)

**S5 Fig. The trend of the ratio of cumulative observed to cumulative expected utilization.** The red lines are for outpatient visits and the blue lines for inpatient discharges.
(PDF)

**S1 STROBE Checklist. Checklist of items that should be included in reports of cohort studies.** Adapted from the STROBE Statement, licensed under CC BY 4.0. Original source: https://www.strobe-statement.org/).
(DOCX)

## Author contributions

**Conceptualization:** Hong Xiao, Yuechong Cui, Joseph M. Unger.

**Data curation:** Hong Xiao, Fang Liu.

**Formal analysis:** Hong Xiao.

**Funding acquisition:** Yuechong Cui, Joseph M. Unger.

**Investigation:** Hong Xiao.

**Methodology:** Hong Xiao, Guannan Bai, Fang Liu, Yuechong Cui, Joseph M. Unger.

**Project administration:** Hong Xiao, Yuechong Cui.

**Resources:** Hong Xiao.

**Software:** Hong Xiao.

**Supervision:** Hong Xiao, Joseph M. Unger.

**Validation:** Hong Xiao, Fang Liu, Joseph M. Unger.

**Visualization:** Hong Xiao, Fang Liu, Joseph M. Unger.

**Writing – original draft:** Hong Xiao.

**Writing – review & editing:** Hong Xiao, Guannan Bai, Fang Liu, Yuechong Cui, Joseph M. Unger.

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
