## [Editor Report · Decision Letter 0]

28 May 2025

Dear Dr Xiao,

Thank you for submitting your manuscript entitled "The Incomplete Road to Recovery: Policy Stringency and Prolonged Impact of the COVID-19 Pandemic on Healthcare Services Utilization in China" for consideration by PLOS Medicine.

Your manuscript has now been evaluated by the PLOS Medicine editorial staff and I am writing to let you know that we would like to send your submission out for external peer review.

For clinical studies, please upload a copy of your trial study protocol as a supporting information file. The study protocol should be the version submitted for approval to the institutional review board or ethics committee, should include any amendments to the study protocol, as well as the date of their approval by the institutional review or ethics committee. Please also detail any deviations from the study protocol in the Methods section of your manuscript. The editors will consider the protocol and study conduct prior to a final decision for external review.

Please re-submit your manuscript within two working days, i.e. by May 30 2025 11:59PM.

Kind regards,

Andreia Cunha, PhD

Senior Editor

PLOS Medicine

---

## [Decision Letter · Decision Letter 1]

9 Sep 2025

Dear Dr Xiao,

Sincere apologies for the delay in getting back to you with a decision, which was due to challenges in securing all the necessary Reviewers. Many thanks for submitting your manuscript "The Incomplete Road to Recovery: Policy Stringency and Prolonged Impact of the COVID-19 Pandemic on Healthcare Services Utilization in China" (PMEDICINE-D-25-01792R1) to PLOS Medicine. The paper has been reviewed by subject experts and a statistician; their comments are included below and can also be accessed here: [LINK]

As you will see, the reviewers find your work of considerable interest but they raise some very important points that we would want to see resolved before consulting them again. After discussing the paper with the editorial team and an academic editor with relevant expertise, I'm pleased to invite you to revise the paper in response to the reviewers' comments. In particular, it seems to us to be essential that the causal language is revied, that the suggested sensitivity analysis are added, and that the concerns regarding the assumptions around lower healthcare utilization and alternative reasons for this are explored further. In addition, it is also our editorial view that a more granular analysis of the available data would be required. We plan to send the revised paper to some or all of the original reviewers, and we cannot provide any guarantees at this stage regarding publication.

We ask that you submit your revision by Dec 09 2025 11:59PM. However, if this deadline is not feasible, please contact me by email, and we can discuss a suitable alternative.

Don't hesitate to contact me directly with any questions (acunha@plos.org).

Best regards,

Andreia

Andreia Cunha, PhD

Senior editor

PLOS Medicine

acunha@plos.org

Comments from the reviewers:

Reviewer #1: The paper presents an interrupted time series analysis of healthcare utilization in China after the onset of the COVID-19 pandemic. It is generally a well written and presented paper. I have the following comments regarding the statistical aspects of the work.

- Some further details on the justification for some modelling choices could be provided. The negative binomial regression model was chosen over a Poisson due to over-dispersion and likelihood ratio tests. Evidence of over-dispersion and the results of these likelihood ratio tests could be provided. Furthermore, the PACF could be shown to justify the use of Newey-West standard errors with three lags. After fitting the proposed model, was there any evidence of residual correlation?

- Confidence intervals were produced using a parametric bootstrap approach. Why was this approach taken over using the asymptotic distribution of the model coefficients?

- The approach assumes a stationary distribution in the training period, that is the period considered before the COVID-19 pandemic. Visual inspection of the data, or of the residuals, could be presented to help to justify this assumption.

- The conclusion that the COVID-19 pandemic impacted healthcare utilisation assumes that there were no other factors that occurred during this time which could have also have affected healthcare utilisation. This could be considered in the discussion when interpreting the results. If there were other factors at play, could these have been included as covariates in the model to account for them?

Reviewer #2: This is a relevant and well-written article examining the impact of the COVID-19 pandemic on outpatient and inpatient care in China. While most studies focus on short-term effects, this analysis provides valuable insights by covering a longer period. It demonstrates that—assuming pre-pandemic secular trend in healthcare utilization—the effects of the pandemic persist at least through 2024, with significant regional variation. Furthermore, the study confirms a clear association between the stringency of public health measures adopted during the emergency and patterns of healthcare utilization.

Although the authors did not disaggregate the results by sex, age group, or health conditions, the article remains valuable for presenting aggregate data on healthcare activity during the pandemic, offering a clear summary of its overall impact on the health system. If these findings were discussed alongside data on healthcare spending, infrastructure, or human resources at the national level, they could yield valuable insights into healthcare productivity during this period—a highly relevant and often debated topic in recent years.

The results support the authors' claims, and the methods used align with those applied in similar studies, with only minor adaptations tailored to the specific context of this research. Particularly valuable are eFigure 1 and eFigure 2, which illustrate healthcare utilization trends throughout the study period, the model fitting for the pre-pandemic period, and counterfactual scenarios at both national and regional levels. With adequate access to the data, the results of the study could be easily reproduced.

I recommend accepting the manuscript with minor revisions.

Line 459 - Consider expanding the color gradient to enhance visual differentiation of the pandemic's impact.

Line 463 - Please enlarge the graphic titles and legends. For graphics C and D, consider inverting the colors, using red to indicate decreases and blue for increases. Ensure this color scheme is clearly described in the figure legend.

Line 473 - Please ensure that all figures are properly numbered (e.g., Figure A1, Figure A2). Also, consider inverting the colors—using red to indicate decreases and blue for increases—and ensure color usage is consistent throughout the article.

Line 484 - Consider removing '02/2024-04/2024: Post Zero-COVID period.' Please enlarge the graphic titles and legends and reduce the size of the data points to better distinguish between color groups.

Reviewer #3: Thanks for giving me the opportunity to review this. interesting paper, but there are some major concerns for this manuscript.

Major concerns:

1. Causal claims exceed what the design supports. The manuscript repeatedly attributes declines and "incomplete recovery" in healthcare utilization to the Zero‑COVID policy (and to policy stringency). The analyses are ecological, aggregate interrupted time‑series with province-level stringency indices. This design cannot establish causality or apportion effects between policy, behavior change, healthcare supply constraints, mortality, demographic shifts, or other contemporaneous system changes. The language in the abstract, results, and discussion is causal/attributive and should be substantially toned down unless stronger causal identification is provided (e.g., credible quasi‑experimental contrasts, instrumental variables, or granular data linking policy actions to supply/demand changes).

The linear regression analysis associating the Policy Stringency Index and utilization is particularly weak as it does not adequately control for time‑varying confounders (e.g., case counts, health system capacity changes, economic shocks, mobility restrictions independent of formal policy index, telemedicine uptake). Without these, the negative association may simply reflect co-occurrence of outbreaks (high cases → high stringency → lower utilization) and not an independent effect of policy.

2. Incomplete control for time‑varying confounding and alternative explanations. The model controls for secular trend, month fixed effects, and Spring Festival days only. It does not sufficiently adjust for the most important time‑varying factors that plausibly affect utilization:

Local COVID incidence (using reliable case estimates) and healthcare burden (e.g., hospital occupancy, ICU strain).

Mobility measures (internal travel data) and transport suspension events.

Supply‑side disruptions: temporary hospital/clinic closures, staff reallocation, suspension of elective services.

Changes in population (excess mortality among older adults) and insurance coverage changes that the Discussion speculates about.

Authors used IHME estimates after official reporting stopped, but do not show sensitivity analyses to alternative case estimates or to omission of case counts. This omission weakens attribution to stringency.

3. Problematic handling and imputation of stringency and missing data. The stringency index is imputed after Feb 28, 2023 by "decrementing 0.3 points per day until reaching zero." This is an arbitrary assumption with no justification and likely misrepresents policy heterogeneity in late 2022-2024. Such imputation could bias the stringency-utilization relationship and regional comparisons.

Missing province-level monthly utilization values were imputed as the mean of adjacent months; December 2022 was imputed from January 2023 due to "abrupt end of Zero‑COVID policy." These ad hoc imputations, especially around a key break point, are not acceptable without sensitivity analyses demonstrating robustness to alternative imputation methods (e.g., multiple imputation, interpolation with uncertainty, or excluding months with irregular reporting).

The authors must provide diagnostics showing the extent and pattern of missingness and justify the imputation strategy statistically.

4. Interpretation of 'recovery' and 'unmet need' is unsupportable. The manuscript treats failure to return to pre‑pandemic utilization volumes as evidence of unmet need and healthcare inequity. This is not necessarily true: utilization may have become more appropriate (reduction of low‑value care), shifted to telemedicine, or been offset by changes in disease incidence (e.g., fewer injuries, less influenza) or demographic changes (excess mortality, declining births). The authors do not provide any outcome or quality data (mortality, emergency admissions for time‑sensitive conditions) to demonstrate harm from decreased utilization. Concluding that health needs are unmet is speculative.

To substantiate unmet need claims, authors would need linkage to health outcomes (excess mortality by cause, cancer stage at diagnosis, avoidable hospitalization rates) or at least present analysis by service type (elective vs. emergency; chronic disease follow‑up vs. preventive care).

5. Over-reliance on aggregate data without subgroup or service‑type analyses. Aggregate, hospital‑level monthly counts mask heterogeneity critical for interpretation. The inability to stratify by age, sex, admission type (elective vs emergency), specialty, or payer limits the value of conclusions about equity and policy implications.

The manuscript should either: (a) obtain and analyze more granular data (if available) to explore these distinctions, or (b) substantially hedge conclusions and present the work strictly as descriptive national/regional trends with clear limitations.

6. Use of case data and IHME estimates. The paper mixes official reported cases (until Nov 30, 2022) with IHME modelled estimates thereafter. The potential biases and uncertainty in IHME estimates—and their impact on analysis—are not discussed. Authors must perform sensitivity checks using alternative sources or explicitly quantify uncertainty introduced by using modeled case counts.

7. Substantially rewrite the claims to avoid causal language unless supported by stronger methods. Reframe the manuscript as descriptive/associational unless authors can provide stronger identification.

Minior comments:

1. Several figures are difficult to interpret (tiny thumbnails, unclear labels, confusing color schemes). The supplemental reviews suggested plotting case counts and utilization on a single graph for illustrative provinces—this would be useful. Figures that drive the main causal assertions (stringency vs utilization) must be clearer and transparent.

Regression outputs (coefficients, IRRs, dispersion) should be presented in main tables, not only narrative text.

2. Numerous typographical and formatting errors throughout (e.g., "ublizabon" etc. in the submitted PDF); these suggest insufficient editorial polish.

3. Some claims in the Discussion (e.g., large declines in insurance enrollment with specific numbers) are referenced to press articles or non‑peer reviewed sources. Prefer peer‑reviewed evidence or official statistics; otherwise clearly label as speculative or anecdotal.

Clarify definitions: what counts as outpatient visits (includes emergency?), inpatient discharges (does this include deaths during admission?), hospital levels included (all hospitals? private clinics?).

4. Explain whether inpatients who died shortly after admission were included; reviewer comments raised this point and it affects interpretation of hospitalization volume.

Clarify the time window used to declare a province "not recovered" (three consecutive months Feb-Apr 2024 below expected). Why three months? Is this robust to choosing two or four months?

---

* Please upload any figures associated with your paper as individual TIF or EPS files with 300dpi resolution at resubmission; please read our figure guidelines for more information on our requirements: http://journals.plos.org/plosmedicine/s/figures. While revising your submission, we strongly recommend that you use PLOS's NAAS tool (https://ngplosjournals.pagemajik.ai/artanalysis) to test your figure files. NAAS can convert your figure files to the TIFF file type and meet basic requirements (such as print size, resolution), or provide you with a report on issues that do not meet our requirements and that NAAS cannot fix.

After uploading your figures to PLOS's NAAS tool - https://ngplosjournals.pagemajik.ai/artanalysis, NAAS will process the files provided and display the results in the "Uploaded Files" section of the page as the processing is complete.

If the uploaded figures meet our requirements (or NAAS is able to fix the files to meet our requirements), the figure will be marked as "fixed" above. If NAAS is unable to fix the files, a red "failed" label will appear above.

When NAAS has confirmed that the figure files meet our requirements, please download the file via the download option, and include these NAAS processed figure files when submitting your revised manuscript.

FIGURES AND TABLES

SUPPLEMENTARY MATERIAL

REFERENCES

OBSERVATIONAL STUDIES

* Abstract: Please include the study design, population and setting, number of participants, years during which the study took place (enrollment and follow up), length of follow up, and main outcome measures.

* Please ensure that the study is reported according to the STROBE (or appropriate STOBE extension) guideline (available from: https://www.equator-network.org/reporting-guidelines/strobe) and include the completed STROBE (or STROBE extension) checklist as Supporting Information. Please add the following statement, or similar, to the Methods: "This study is reported as per the Strengthening the Reporting of Observational Studies in Epidemiology (STROBE) guideline (S1 Checklist)." When completing the checklist, please use section and paragraph numbers, rather than page numbers.

* For all observational studies, in the manuscript text, please indicate: (1) the specific hypotheses you intended to test, (2) the analytical methods by which you planned to test them, (3) the analyses you actually performed, and (4) when reported analyses differ from those that were planned, transparent explanations for differences that affect the reliability of the study's results. If a reported analysis was performed based on an interesting but unanticipated pattern in the data, please be clear that the analysis was data driven.

* Please state in the Methods section whether the study had a prospective protocol or analysis plan. If a prospective analysis plan (from your funding proposal, IRB or other ethics committee submission, study protocol, or other planning document written before analyzing the data) was used in designing the study, please include the relevant document(s) with your revised manuscript as a Supporting Information file to be published alongside your study and cite it in the Methods section. A legend for this file should be included at the end of your manuscript. If no such document exists, please make sure that the Methods section transparently describes when analyses were planned, and when/why any data-driven changes to analyses took place. Changes in the analysis, including those made in response to peer review comments, should be identified as such in the Methods section of the paper, with rationale.

MODELLING STUDIES

The following list is derived from Geoffrey P Garnett, Simon Cousens, Timothy B Hallett, Richard Steketee, Neff Walker. Mathematical models in the evaluation of health programmes. (2011) Lancet DOI:10.1016/S0140-6736(10)61505-X:

* If pertinent, please provide a diagram that shows the model structure, including how the natural history of the disease is represented, the process and determinants of disease acquisition, and how the putative intervention could affect the system.

* Please provide a complete list of model parameters, including clear and precise descriptions of the meaning of each parameter, together with the values or ranges for each, with justification or the primary source cited and important caveats about the use of these values noted.

* Please provide a clear statement about how the model was fitted to the data, including goodness-of-fit measure, the numerical algorithm used, which parameter varied, constraints imposed on parameter values, and starting conditions.

* For uncertainty analyses, please state the sources of uncertainties quantified and not quantified [can include parameter, data, and model structure].

* Please provide sensitivity analyses to identify which parameter values are most important in the model. Uncertainty estimates seek to derive a range of credible results on the basis of an exploration of the range of reasonable parameter values. The choice of method should be presented and justified.

* Please discuss the scientific rationale for the choice of model structure and identify points where this choice could influence conclusions drawn. Please also describe the strength of the scientific basis underlying the key model assumptions.

---

## [Decision Letter · Decision Letter 2]

30 Jan 2026

Dear Dr. Xiao,

Sincere apologies for the delay in getting back to you with a decision and thank you very much for re-submitting your manuscript "The Incomplete Road to Recovery: Policy Stringency and Prolonged Impact of the COVID-19 Pandemic on Healthcare Services Utilization in China" (PMEDICINE-D-25-01792R2) for review by PLOS Medicine.

I have discussed the paper with my colleagues and the academic editor and it was also seen again by two of the original reviewers. I am pleased to say that provided the remaining reviewer, editorial and production issues are dealt with we are planning to accept the paper for publication in the journal.

[LINK]

We look forward to receiving the revised manuscript by Feb 06 2026 11:59PM.

Sincerely,

Andreia Cunha, PhD

Senior Editor

PLOS Medicine

plosmedicine.org

Requests from Editors:

GENERAL EDITORIAL REQUESTS

* Please revise your title to follow PLOS Medicine's style. Your title must be nondeclarative and not a question. It should begin with main concept if possible. "Effect of" should be used only if causality can be inferred, i.e., for an RCT. Please place the study design ("A randomized controlled trial," "A retrospective study," "A modelling study," etc.) in the subtitle (ie, after a colon).

* Please confirm that your abstract complies with our requirements, including format (three sections: Background, Methods and Findings, and Conclusions) and providing all the information relevant to this study type https://journals.plos.org/plosmedicine/s/submission-guidelines#loc-abstract

* Please ensure that the Introduction ends with a clear description of the study question or hypothesis.

* Please ensure that all abbreviations are defined at first use throughout the text.

* Please confirm that all numbers presented in the abstract are present and identical to numbers presented in the main manuscript text.

GENERAL

* Please review your text for claims of novelty or primacy (e.g. 'for the first time') and remove this language. In addition, please check that any use of statistical terms (such as trend or significant) are supported by the data, and if not please remove them.

* Please remove the 'conclusions' subheading from the discussion. Please also remove any other subheadings from the discussion.

* Statistical reporting: Please revise throughout the manuscript, including tables and figures.

- Please report statistical information as follows to improve clarity for the reader ""22% (95% CI [13,28]; p</=)"".

- Please separate upper and lower bounds with commas instead of hyphens as the latter can be confused with reporting of negative values.

- Please repeat statistical definitions (HR, CI etc.) for each set of parentheses."

ABSTRACT

* In the last sentence of the Methods and Findings section, please describe the main limitation(s) of the study's methodology.

* Please also include the important dependent variables that are adjusted for in the analyses.

* Please revise the last sentence of the Background to active voice (e.g. We comprehensively analyzed...).

* Please also define what the policy stringency index is in the Abstract for the general reader.

* In the author summary, in the final bullet point of 'What Do These Findings Mean?', please include the main limitations of the study in non-technical language.

FUNDING STATEMENT

* The funding statement should include: specific grant numbers, initials of authors who received each award, URLs to sponsors’ websites. Also, please state whether any sponsors or funders (other than the named authors) played any role in study design, data collection and analysis, the decision to publish, or preparation of the manuscript. If they had no role in the research, include this sentence: “The funders had no role in study design, data collection and analysis, decision to publish, or preparation of the manuscript.”

* It appears that one or more study authors is affiliated with one or more of the agencies that funded the study. Thus, the statement “The funders had no role in study design, data collection and analysis, decision to publish, or preparation of the manuscript” does not apply. Please revise the Financial Disclosure accordingly, as in "[Author name] is [author's role] at [funding agency]. The funders had no other role in study design…..”

COMPETING INTERESTS STATEMENT

* All authors must declare their relevant competing interests per the PLOS policy, which can be seen here: https://journals.plos.org/plosmedicine/s/competing-interests For authors with ties to industry, please indicate whether any of the interests has a financial stake in the results of the current study.

DATA AVAILABILITY

* PLOS Medicine requires that the de-identified data underlying the specific results in a published article be made available, without restrictions on access, in a public repository or as Supporting Information at the time of article publication, provided it is legal and ethical to do so. Please see the policy at

http://journals.plos.org/plosmedicine/s/data-availability

and FAQs at

http://journals.plos.org/plosmedicine/s/data-availability#loc-faqs-for-data-policy

* The Data Availability Statement (DAS) requires revision. For each data source used in your study:

FIGURES

* Please provide titles and legends for all figures and tables (including those in Supporting Information files). Please define all acronyms used in each figure or table in its corresponding legend.

* Please ensure that where relevant figures include 95% CIs.

* Please consider avoiding the use of red and green in order to make your figure more accessible

* Please confirm that the appropriate usage rights apply to the use of this map. Please see our guidelines for map images: https://journals.plos.org/plosmedicine/s/figures#loc-maps

OBSERVATIONAL, COHORT, CROSS-SECTIONAL, AND CASE CONTROL STUDIES

* Please ensure that the study is reported according to the STROBE guideline, and include the completed STROBE checklist as Supporting Information. Please add the following statement, or similar, to the Methods: ""This study is reported as per the Strengthening the Reporting of Observational Studies in Epidemiology (STROBE) guideline (S1 Checklist).""

When completing the checklist, please use section and paragraph numbers, rather than page numbers."

* Did your study have a prospective protocol or analysis plan? Please state this (either way) early in the Methods section.

c) In either case, changes in the analysis-- including those made in response to peer review comments-- should be identified as such in the Methods section of the paper, with rationale."

* Your study is observational and therefore causality cannot be inferred. Please remove language that implies causality and refer to associations instead.

* For all observational studies, in the manuscript text, please indicate: (1) the specific hypotheses you intended to test, (2) the analytical methods by which you planned to test them, (3) the analyses you actually performed, and (4) when reported analyses differ from those that were planned, transparent explanations for differences that affect the reliability of the study's results. If a reported analysis was performed based on an interesting but unanticipated pattern in the data, please be clear that the analysis was data-driven.

Comments from Reviewers:

Reviewer #1: I thank the authors for their response, which have generally covered my previous comments. It would be good to include, perhaps in the supplementary, the results which led to the discussed modelling choices. In line 192, a p-value is reported justifying the use of negative binomial regression, but it is not clear if this p-value is using the aggregated national data, or a selected province-level dataset. In addition, the PACFs and Cumby-Huizings results could be shown for each model fit (i.e., at the province-level) to support the chosen lags in the Newey-West standard errors.

Reviewer #2: This is a relevant and well-written article that summarizes the impact of the COVID-19 pandemic on outpatient and inpatient care. The period analyzed in this article is longer than in previous articles on this topic. It demonstrates that—assuming pre-pandemic secular trends in healthcare utilization—the effects of the pandemic persist at least through 2024, with regional variations. The manuscript has value for the global health literature because the case of China was highly particular, and the methods can be reproduced in other countries.

I recommend accepting the article without additional revisions.

[LINK]

---

## [Editor Report · Decision Letter 3]

5 Mar 2026

Dear Dr Xiao,

On behalf of my colleagues and the Academic Editor, Margaret Kruk, I am pleased to inform you that we have agreed to publish your manuscript "Policy stringency during the COVID-19 pandemic and healthcare services utilization in China: an interrupted time-series analysis" (PMEDICINE-D-25-01792R3) in PLOS Medicine.

Before your manuscript can be formally accepted you will need to complete some formatting changes, which you will receive in a follow up email. For example, it will include edits to the author summary to improve readability and avoid repetition. Please be aware that it may take several days for you to receive this email; during this time no action is required by you. Once you have received these formatting requests, please note that your manuscript will not be scheduled for publication until you have made the required changes.

PRESS

Sincerely,

Andreia Cunha, PhD

Senior Editor

PLOS Medicine